# Genome-Wide Analysis of the Temporal Genetic Changes in *Streptococcus pneumoniae* Isolates of Genotype ST320 and Serotype 19A from South Korea

**DOI:** 10.3390/microorganisms9040795

**Published:** 2021-04-10

**Authors:** Jin Yang Baek, Sun Ju Kim, Juyoun Shin, Yeun-Jun Chung, Cheol-In Kang, Doo Ryeon Chung, Jae-Hoon Song, Kwan Soo Ko

**Affiliations:** 1Asia Pacific Foundation for Infectious Diseases (APFID), Seoul 06351, Korea; jy34.baek@gmail.com (J.Y.B.); dr.chung@samsung.com (D.R.C.); ansorp@gmail.com (J.-H.S.); 2Department of Microbiology and Samsung Medical Center, Sungkyunkwan University School of Medicine, Suwon 16419, Korea; n.e.coli.1822@gmail.com; 3Department of Microbiology, College of Medicine, The Catholic University of Korea, Seoul 03083, Korea; sjy83ww@gmail.com (J.S.); yejun@catholic.ac.kr (Y.-J.C.); 4Precision Medicine Research Center, Integrated Research Center for Genome Polymorphism, College of Medicine, The Catholic University of Korea, Seoul 03083, Korea; 5Division of Infectious Diseases, Samsung Medical Center, Sungkyunkwan University School of Medicine, Seoul 06351, Korea; collacin@gmail.com

**Keywords:** pneumococci, recombination, whole-genome sequencing, serotype 19A, ST320

## Abstract

Since the introduction of the pneumococcal conjugate vaccine, an increase in the incidence of *Streptococcus pneumoniae* serotype 19A and sequence type 320 (19A-ST320) isolates have been observed worldwide including in South Korea. We conducted a genome-wide analysis to investigate the temporal genetic changes in 26 penicillin-non-susceptible 19A-ST320 pneumococcal isolates from a hospital in South Korea over a period of 17 years (1999; 2004 to 2015). Although the strains were isolated from a single hospital and showed the same genotype and serotype, a whole-genome sequencing (WGS) analysis revealed that the *S. pneumoniae* isolates showed more extensive genetic variations compared with a reference isolate obtained in 1999. A phylogenetic analysis based on single nucleotide polymorphisms (SNPs) showed that the pneumococcal isolates from South Korea were not grouped together into limited clusters among the 19A-ST320 isolates from several continents. It was predicted that recombination events occurred in 11 isolates; larger numbers of SNPs were found within recombination blocks compared with point mutations identified in five isolates. WGS data indicated that *S. pneumoniae* 19A-ST320 isolates might have been introduced into South Korea from various other countries. In addition, it was revealed that recombination may play a great role in the evolution of pneumococci even in very limited places and periods.

## 1. Introduction

The pathogen *Streptococcus pneumoniae* causes severe infections worldwide especially in infants, young children and the elderly. The polysaccharide capsule of *S. pneumoniae* is considered to be the most important pneumococcal virulence factor [1]. Although more than 90 serotypes have been identified in *S. pneumoniae*, relatively few of them are commonly associated with invasive pneumococcal diseases (IPDs) [1]. The 7-valent pneumococcal conjugate vaccine (PCV7), which targets seven serotypes (4, 6B, 9V, 14, 18C, 19F and 23F), was introduced in 2000 for immunizing children [2]. It has been replaced by PCV13, which targets six additional serotypes (1, 5, 7F, 3, 6A and 19A) and has also been approved for the prevention of IPDs and non-invasive pneumonia in adults [3]. Since the introduction of pneumococcal vaccines, the incidence of IPDs has decreased in all age groups worldwide [4]. 

The use of PCVs has led to a change in the serotype distribution among pneumococcal isolates [1]. After the introduction of PCV7, an increase in the occurrence of serotype 19A was particularly evident [5], which was partly due to “serotype switching” or “capsular switch” [6]. Although *S. pneumoniae* 19A-ST320 isolates had been reported before the introduction of PCV7 in South Korea [7], serotype 19A remained the most prevalent serotype in South Korea even after the introduction of PCV13 [8]. Among the *S. pneumoniae* 19A isolates, ST320 was the dominant clone in Asian countries including South Korea [8,9]. The *S. pneumoniae* 19A-ST320 isolates have shown multidrug resistance including penicillin resistance [7,9].

*S. pneumoniae* exhibits a high degree of genomic plasticity as evidenced by the degree of genomic variability between isolates; it was reported that approximately 74% of the genome is shared in common between strains as a pan-genome [10]. Whole-genome sequencing (WGS) facilitates high-resolution strain typing and can effectively identify recombination and genomic variability. WGS studies have revealed that *S. pneumoniae* rapidly and successfully develops various characteristics including antibiotic resistance via recombination and point mutations [6,11,12]. However, the extent to which recombination plays an important role in the evolution of pneumococci even in isolates from limited areas remains unknown. 

In this study, we performed a genome-wide analysis to examine the temporal genetic changes in *S. pneumoniae* 19A-ST320 isolates during a period of 17 years using isolates obtained from a single tertiary-care hospital in South Korea. Our study shows that *S. pneumoniae* 19A-ST320 isolates have evolved dynamically via frequent point mutations and recombination. 

## 2. Materials and Methods

### 2.1. Bacterial Strains and In Vitro Induction of Tigecycline-Resistant Mutants

A total of 26 *S. pneumoniae* isolates (two isolates per year (1999; 2004 to 2015)) were obtained from a single tertiary-care hospital in South Korea (Samsung Medical Center, Seoul, Korea) and analyzed (Table 1). They were selected from 156 isolates obtained in the institution during the study period, based on the criteria of serotype (19A), genotype (ST320) and penicillin non-susceptibility (minimum inhibitory concentration [MIC] ≥ 2 mg/L).

### 2.2. Serotyping and Genotyping

Serotyping of the pneumococcal isolates was determined via the capsular Quellung method using commercial antisera (Statens Serum Institut, Copenhagen, Denmark), according to the manufacturer’s instructions. The genotypes of the isolates were determined by performing multilocus sequence typing [13].

### 2.3. Antimicrobial Susceptibility Test

Antimicrobial susceptibility tests of *S. pneumoniae* isolates were performed via the broth microdilution method according to the Clinical and Laboratory Standards Institute (CLSI) [14]. *S. pneumoniae* ATCC 49619 and *Staphylococcus aureus* ATCC 29213 were used as the control strains. 

### 2.4. Genome Sequencing and Assembly

*S. pneumoniae* 99-176, which was isolated in 1999, was sequenced de novo by Macrogen (Seoul, South Korea) using a PacBio RS II system (Pacific Biosciences, Menlo Park, CA, USA) after which a 20 kb single molecule real-time (SMRT) bell template library (Pacific Biosciences) was constructed. Open reading frames were predicted and annotated using Prokka (v1.13). Each predicted protein was compared against a protein database using a basic local alignment search tool for proteins (BLASTP) with a minimum cut-off of 30% identity and 80% coverage. The annotated sequence of *S. pneumoniae* 99-176 was deposited in the GenBank nucleotide sequence database under accession number CP063829. 

To identify the genetic differences between *S. pneumoniae* 99-176 and the other strains, the other 25 19A-ST320 isolates were sequenced using the Illumina HiSeq 2000 Preliminary Performance Parameters (151 bp paired-end reads). The sequences of the other 25 isolates covered 99.4% to 100% of those of the strain 99-176. The reads were deposited in the NCBI Sequence Read Archive under the GenBank accession number PRJNA671606.

### 2.5. Prediction of Recombination Sites via Phylogenetic Analysis

To map the reads to the reference, the trimmed reads for the 178 *S. pneumoniae* 19A-ST320 isolates, including the 25 isolates identified in this study, were aligned against the reference genome of 99-176 using Snippy version 4.4.5 (https://github.com/tseemann/snippy/, accessed on 12 February 2021). The quality of the SNPs was checked by a threshold of 0.9 and a depth over 30 using Snippy. The whole-genome sequences of *S. pneumoniae* 19A-ST320 isolates, which were classified as global pneumococcal sequence cluster 1 (GPSC1), were retrieved from the website of The Global Pneumococcal Sequencing (GPS) Project (https://www.pneumogen.net/gps/, accessed on 12 February 2021) [15]; sequences of SP61 and SP64 were also included in the analysis. The whole-genome sequences of 178 strains were used to just call SNPs and were not assembled. The whole-genome core SNP alignment output from Snippy was used for a downstream phylogenetic analysis. A maximum-likelihood tree was generated using RAxML version 8.2.10 based on the SNPs with a minimum fraction of 90% among ≥20 reads [16]. To identify the regions of genetic recombination, we used Gubbins (v.2.3.4), which uses an algorithm to iteratively identify the loci containing elevated densities of base substitutions while concurrently constructing a phylogeny based on the putative point mutations outside of these regions [17]. The resulting phylogenetic tree, isolate metadata, core genome SNPs and recombination sites were visualized using Phandango version 1.3.0 [18].

## 3. Results

### 3.1. WGS Statistics

The genome of *S. pneumoniae* 99-176, which was determined de novo as a reference and was assembled de novo by method of HGAP3 and Quiver (Pacific Biosciences, Menlo Park, CA, USA), was found to be 2,089,994 bp in length, containing 2064 coding sequences as well as 58 tRNA and 12 rRNA genes. The sequencing statistics of the other *S. pneumoniae* isolates are shown in Appendix A. The numbers of contigs of the isolates analyzed in this study, except for isolate 15-019, ranged from 22 to 30. A total of 237 contigs were obtained and analyzed from isolate 15-019. The reads obtained by Illumina were assembled by SPAdes v3.11.1 (http://cab.spbu.ru/software/spades/, accessed on 12 February 2021). The assembled sequence lengths of the 24 isolates except for 15-019 ranged from 2,026,658 to 2,083,098 bp, which covered 97.0% to 99.7% of the genome of isolate 99-176. The depth of coverage on the reference sequence of 99-176 was 757.164 on average, ranging from 495.246 to 1753.27. The overall G+C contents of 26 isolates were 39.52 to 39.84%.

### 3.2. SNPs

The SNPs of the isolates were determined via a comparison of their Illumina reads with whole-genome sequences of isolate 99-176, which were generated de novo (Table 1). As the sequences of isolate 15-019 showed a larger number of SNPs (11,542) compared with those of the other *S. pneumoniae* isolates included in this study, we analyzed the SNPs of 15-019 separately. The number of SNPs present in the genomes of the other 24 isolates excluding 15-019 ranged from 7 (99-192) to 2764 (07-093). The number of insertions or deletions (INDELs) were found to range from 1 to 49 in the 24 isolates, compared with the genome of isolate 99-176. Isolate 15-019 showed 90 INDELs. Based on a BLAST analysis, the genome of isolate 15-019 was the most similar to those of *S. pneumoniae* strains SP61 and SP64 isolated in Germany (https://www.ncbi.nlm.nih.gov/nuccore/CP018137.1/, accessed on 12 February 2021), which also showed serotype 19A and genotype ST320 [19]; using Snippy (v4.4.5), the number of SNPs between 15-019 and SP61 and SP64 were estimated to be 53 and 33, respectively.

Figure 1 shows the distribution of genes containing sequence variations. Although the rates of SNPs were less than 1% in most genes of most *S. pneumoniae* isolates, high genetic variations were found in certain regions of a few isolates. Genes exhibiting SNP rates of ≥4% were found in seven isolates (Figure 1). Particularly, several regions including genes containing SNP rates of ≥4% were identified in isolates 07-093 and 14-109. It was found that isolate 15-019 contained more highly variable genes compared with the other isolates.

### 3.3. Phylogenetic Analyses and Prediction of Recombination Sites

We analyzed the whole-genome sequences of 179 *S. pneumoniae* 19A-ST320 isolates including 153 isolates that belonged to GPSC1 [15].15 A phylogenetic tree based on the SNPs showed multichotomous branching with several obvious sub-clusters (Figure 2A). In the phylogenetic tree, the *S. pneumoniae* isolates from South Korea were not grouped into limited clusters among the isolates from several continents. Although pairs of isolates 08-114 and 12-102, 11-194 and 14-212, 04-177 and 14-109 and 99-176 and 99-192 were grouped together, the isolates from South Korea were scattered in the phylogenetic tree, unlike those from China. Several isolates from South Korea showed close relationships with the pneumococcal strains from other continents rather than Asia.

A sub-cluster including isolate 15-019, which showed markedly higher variation compared with other *S. pneumoniae* isolates from South Korea, is shown in Figure 2B. The sub-cluster consisted mainly of isolates from China. Two isolates from Slovenia and one isolate from Germany (SP61) were included in the sub-cluster along with two Korean isolates. Although more variable genes were identified in the whole-genome sequence of 15-019 than in that of 99-176 (a reference strain), only one putative recombination site containing a prophage gene was identified.

We next analyzed only the whole-genome sequences of the *S. pneumoniae* isolates from South Korea that were isolated in this study, excluding 15-019 (Figure 3). In the phylogenetic tree, isolates from the same year were not grouped together except for the isolates obtained in 1999 and no distinct clusters were identified. We predicted recombination sites in the genomes of the isolates using Gubbins (v.2.3.4) (Table 1; Figure 3). Recombination sites predicted using only isolates from Korea were nearly similar to those predicted using 279 *S. pneumoniae* isolates. It was predicted that recombination events occurred in 11 isolates. In the 11 *S. pneumoniae* isolates, 1 (04-177, 11-138 and 14-212) to 11 (07-093 and 09-125) putative recombination blocks were identified. In five isolates (11-194, 05-384, 09-125, 07-093 and 14-109), multiple genes that are presented at the top or bottom in Figure 3, were included in recombination blocks. In the five isolates, the r/m values (the number of SNPs within recombination blocks/the number of SNPs outside recombination blocks) were high; 12.98 for 09-125 and 58.33 for 07-093 (Table 1). The predicted recombination events were not associated with phylogenetic grouping.

## 4. Discussion

To examine the temporal genetic changes in penicillin-non-susceptible *S. pneumoniae* 19A-ST320 isolates, which have spread worldwide including to South Korea after the introduction of PCV7, we performed WGS for two randomly selected isolates per year from a hospital in South Korea. Despite the fact that these isolates were obtained from a single hospital and all showed the same serotype and genotype, the degree of genetic variation was much larger than expected. 

By performing a phylogenetic analysis based on SNPs, we found that the isolates from South Korea did not form one or a few clades among the *S. pneumoniae* 19A-ST320 isolates from several continents. However, certain pneumococcal isolates from South Korea such as 06-300 and 19-019 were clustered with the isolates from China; several isolates were dispersed in the phylogenetic tree. This suggested that all 19A-ST320 isolates, whose incidence increased after the introduction of PCV7, did not disseminate in South Korea via clonal spreading. Instead, the 19A-ST320 pneumococcal isolates from South Korea may have multiple origins. Since the introduction of the vaccine, pneumococcal strains may have been steadily introduced into South Korea from foreign countries. Although the two 19A-ST320 isolates obtained in 1999 have nearly identical genomes, the possibility cannot be excluded that various 19A-ST320 strains existed even before the introduction of the vaccine and have evolved into diverse isolates via point mutations and recombination.

The characteristics of isolate 15-019 are noteworthy. The isolate showed a higher sequence variation than other pneumococcal isolates from South Korea; therefore, it was suspected that frequent genetic recombination events had taken place in 15-019. However, only one recombination in a small region was predicted to occur in isolate 15-019. Therefore, it is speculated that 15-019 or its related strains might have been introduced from a foreign country; for example, from China. In addition, isolate 06-300 was clustered together with isolate 15-019 despite the large genetic variation between the two isolates. In the genome of 06-300, no predicted recombination site was identified. Therefore, there may be no direct association between the two isolates; they or their ancestors might have been imported from different countries.

The distribution of variable genes and recombination sites showed that both point mutations and recombination might have contributed to the temporal genetic changes in *S. pneumoniae* 19A-ST320 isolates during the study period of 17 years. No recombination sites were predicted in 15 pneumococcal isolates from South Korea; however, the genomes of five isolates showed more SNPs within recombination blocks than outside recombination blocks. There were significant differences in the levels of genetic recombination between pneumococcal lineages although hotspots for recombination such as regions including antibiotic resistance genes showed consistency across lineages due to a common selective pressure [20,21]. Our data suggested that the frequency of recombination might have been different between pneumococcal isolates even if they belonged to the same lineage. The introduction of foreign DNA from the environment into the chromosome via transformation and homologous recombination is known to contribute more towards genetic variation than mutations [15]. It has been estimated that nearly 90% of all genetic variations in *S. pneumoniae* have been introduced via recombination [6]. Although only isolates from a single hospital were analyzed, our WGS data also indicated that recombination might play a great role in the evolution of pneumococci.

Three recombination sites were found in multiple isolates from South Korea (Figure 3). As the regions including recombination sites were not identical and the isolates containing the regions were not clustered together, recombination events could not have occurred in their common ancestors. Therefore, all recombination sites identified in this study might have occurred independently. In the analysis including the 19A-ST320 pneumococcal isolates of GPSC1, ancestral recombination was found to be associated with limited isolates from North America. This indicated that recombination in the 19A-ST320 lineage might occur frequently and independently. 

Various genes were included in recombination blocks. First, genes involved in metabolism were identified at the site of recombination. The genes *purL* to *purK* involved in nucleotide biosynthesis and *mnm* genes associated with 2-thiouridine biosynthesis were found in recombination sites of 11-194 [22,23]. The *mnm* genes were also observed in a recombination site of 14-109. The genes required for maltose metabolism, *malP* to *malR* [24], and the genes regulating membrane fluidity, *desR* and *desk* [25], were included in recombination blocks of 14-109. The genes associated with the regulation of glutamine and glutamate metabolism in *S. pneumoniae*, i.e., *glnR* and *glnA* [26], were found in the recombination block of 05-384. In addition, *rsmA* and *prmA*, which encode methyltransferase, were found in the recombination site of 05-384. 

A few genes known to be associated with virulence in *S. pneumoniae* were also identified within recombination blocks. The gene *lytB_1* was found in the recombination sites of 14-109 and 11-194 [27]. The gene *strH*, which was found in a recombination block of 05-384, encodes exo-beta-D-N-acetylglucosaminidase, a known virulence factor in *S. pneumoniae* that promotes resistance to opsonophagocytic killing by human neutrophils [28].

In 14-109, the genes associated with antibiotic resistance were identified in recombination sites; *sulA* associated with sulfonamide resistance [29], several genes encoding enzymes involved in the peptidoglycan biosynthetic pathway and penicillin resistance such as mur genes [30] and a gene associated with vancomycin resistance in *Staphylococcus aureus*, *graS* [31].

Genetic alterations due to the recombination of penicillin-binding proteins encoded by *pbp1a*, *pbp2x* and *pbp2b* are known to contribute to penicillin resistance [32]. However, we could not identify the recombination of regions that contained *pbp* genes because we selected only penicillin-non-susceptible isolates in this study. Although the roles of the genes identified within recombination blocks are unknown, it is probable that recombination results in a variety of traits in the same clone. 

In summary, we performed WGS for two randomly selected *S. pneumoniae* 19A-ST320 isolates per year over 17 years (1999; 2004 to 2015) from one hospital in South Korea. Despite showing the same serotype and genotype, a high degree of genetic variation was observed. A phylogenetic analysis of isolates including 19A-ST320 isolates of GSPC1 showed that several isolates from South Korea were dispersed in the constructed phylogenetic tree, suggesting that the 19A-ST320 isolates from South Korea may have multiple origins. Frequent recombination events were identified in certain isolates, indicating that recombination may play a great role in the evolution of pneumococci although the isolates were obtained from a single hospital.

## Figures and Tables

**Figure 1 microorganisms-09-00795-f001:**
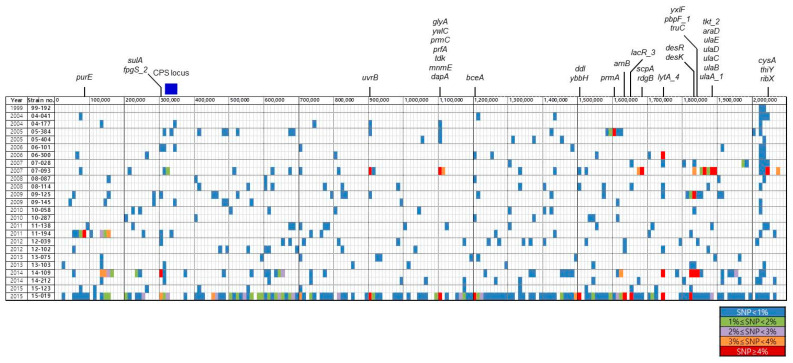
Distribution of genes showing sequence variation. The colored blocks indicate the genes showing sequence variations compared with isolate 99-176. It is represented regardless of the length of the genes. The color was marked differently depending on the degree of variation. The genes showing SNP rates ≥ 4% are shown at the top of the figure. The location of the CPS locus is indicated by a blue box. SNP, single nucleotide polymorphism; CPS, capsular polysaccharide.

**Figure 2 microorganisms-09-00795-f002:**
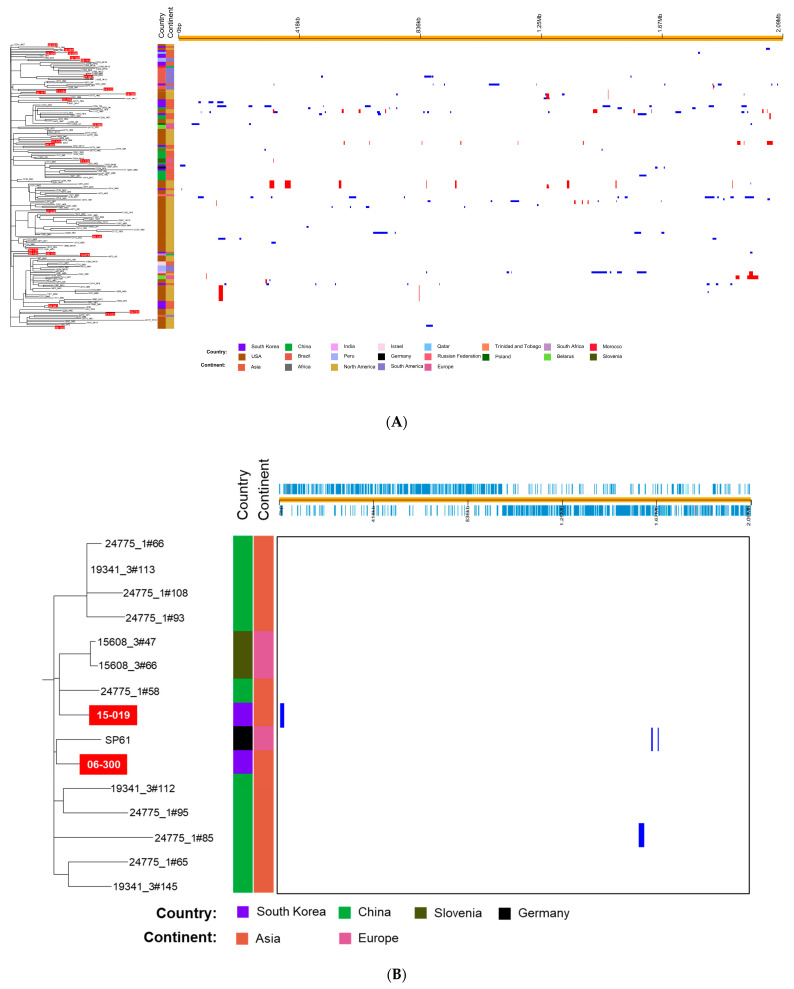
(**A**) Phylogenetic tree and predicted recombination sites of 179 *S. pneumoniae* 19A-ST320 isolates based on whole-genome sequence data of 26 isolates determined in this study. Sequences of 151 isolates were retrieved from the Global Pneumococcal Sequencing (GPS) project (https://www.pneumogen.net/gps/) (accessed on 12 February 2021) and sequences of SP61 and SP64 were obtained from Germany. On the left, the phylogenetic tree constructed via RAxML based on SNPs is shown. The isolates from South Korea are represented with a red background. On the right, recombination sites are shown. Red bars indicate that recombination is ancestral, i.e., occurred at a non-terminal nod. Blue bars are shown if they appeared in one isolate. (**B**) A sub-cluster containing *S. pneumoniae* isolates 15-075 and 06-300, which are represented with a red background. GPS, global pneumococcal sequencing; RAxML, Randomized Axelerated Maximum Likelihood; SNPs, single nucleotide polymorphisms.

**Figure 3 microorganisms-09-00795-f003:**
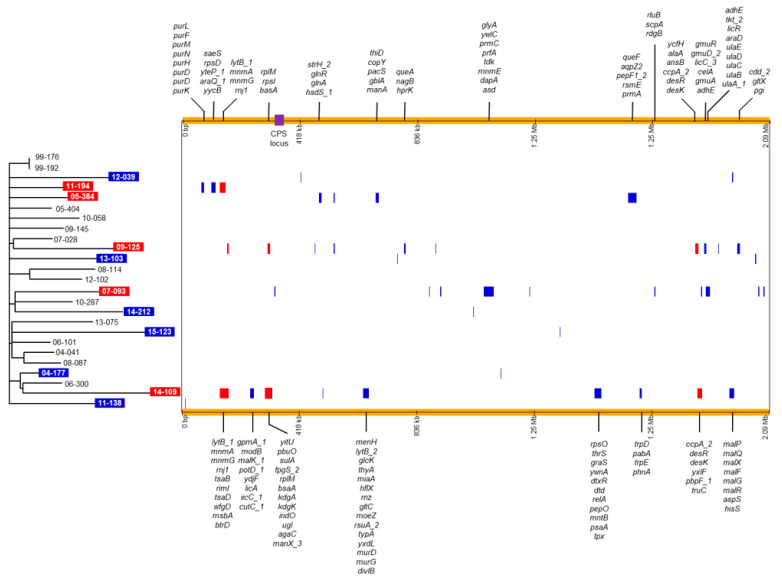
Phylogenetic tree and predicted recombination sites of 25 *S. pneumoniae* 19A-ST320 isolates analyzed in this study. The phylogenetic tree constructed via RAxML based on SNPs is shown on the left. The isolates showing the predicted recombination events are indicated with a colored background; red background indicates that the r/m value was 10 or more; blue background indicates that the r/m value was <1. A horizontal and colored bar showing the predicted recombination events is displayed for each isolate on the right. Recombination blocks predicted in multiple isolates are presented in a red color and those predicted in one isolate are presented in blue. The genes within recombination blocks including multiple genes are shown at the top and bottom of the figure. The location of the CPS locus is indicated by a purple box. RAxML, Randomized Axelerated Maximum Likelihood; SNPs, single nucleotide polymorphisms; CPS, capsular polysaccharide.

**Table 1 microorganisms-09-00795-t001:** Characteristics of the *S. pneumoniae* isolates analyzed in this study.

Isolate No.	Isolation Year	Specimen	MIC (mg/L) ^a^					No of SNPs ^b^	No of INDELs ^b^	Putative Recombination Events	r/m ^c^
			PEN	CRO	LEV	MFX	GFX				
99-176	1999	Sputum	2	1	1	0.12	<0.03	-	-	-	-
99-192	1999	Sputum	4	1	1	0.12	<0.03	7	1	0	0
04-041	2004	Nasopharynx	2	2	1	0.12	<0.03	34	6	0	0
04-177	2004	Blood	4	1	1	0.12	<0.03	53	11	1	0.86
05-384	2005	Sputum	4	2	1	0.12	<0.03	615	17	4	25.56
05-404	2005	Tracheal aspirate	8	8	1	0.12	<0.03	44	7	0	0
06-101	2006	Sputum	4	1	1	0.12	<0.03	37	7	0	0
06-300	2006	Sputum	2	2	1	0.12	<0.03	101	3	0	0
07-028	2007	Sputum	4	1	1	0.12	<0.03	190	11	0	0
07-093	2007	Pus	4	4	0.5	0.06	<0.03	2764	26	11	58.33
08-087	2008	Ear	4	1	1	0.12	<0.03	44	5	0	0
08-114	2008	Sputum	4	2	1	0.12	<0.03	54	10	0	0
09-125	2009	Nasopharynx	4	2	1	0.12	<0.03	479	13	11	12.98
09-145	2009	Sputum	2	1	1	0.12	<0.03	44	8	0	0
10-058	2010	Sputum	4	2	1	0.12	<0.03	51	10	0	0
10-287	2010	Sputum	4	2	1	0.12	<0.03	42	5	0	0
11-138	2011	Nasopharynx	4	1	1	0.12	<0.03	76	9	1	0.43
11-194	2011	Tracheal aspirate	4	1	0.5	0.06	<0.03	748	23	3	37.50
12-039	2012	Blood	4	2	0.5	0.12	<0.03	78	9	2	0.30
12-102	2012	Ear	4	2	1	0.12	<0.03	67	7	0	0
13-075	2013	Pleural fluid	2	1	1	0.12	<0.03	62	9	0	0
13-103	2013	Blood	2	1	1	0.12	<0.03	103	14	2	0.43
14-109	2014	Sputum	4	1	1	0.12	<0.03	2002	49	9	34.77
14-212	2014	Sputum	4	1	1	0.12	<0.03	68	14	1	0.16
15-123	2015	Others	2	1	1	0.12	<0.03	86	13	2	0.19
15-019	2015	Blood	2	1	1	0.12	<0.03	11,542	90	1	ND

^a^ MIC, minimum inhibitory concentration; PEN, penicillin; CRO, ceftriaxone; LEV, levofloxacin; MFX, moxifloxacin; GFX, gemifloxacin; ND, not determined. ^b^ The numbers of single nucleotide polymorphisms (SNPs) and insertion-deletions (INDELs) were measured via a comparison with the nucleotide sequences of isolate 99-176. ^c^ r/m, number of SNPs within recombination blocks/number of SNPs outside recombination blocks.

## Data Availability

The data presented in this study are available on request from the corresponding author.

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
