# Peer review of "Genome-Wide Analysis of the Temporal Genetic Changes in Streptococcus pneumoniae Isolates of Genotype ST320 and Serotype 19A from South Korea"

_microorganisms, 2021, doi:10.3390/microorganisms9040795_

Round 1
Reviewer 1 Report
This paper uses “genome-wide analysis to investigate the temporal genetic changes in 26 penicillin-nonsusceptible 19A-ST320 pneumococcal isolates from a hospital in South Korea over a period of 17 years (1999; 2004 to 2015).” The authors determined that the South Korean hospital samples do not cluster but are rather dispersed throughout the tree showing there is no clonal evolution within South Korea but rather that there was either a pool of variable 19A-ST320 strains in South Korea before 1999 or that strains have been introduced into South Korea over time or a combination of both. While recombination events were far from ubiquitous in the 26 samples, the authors showed that these events injected significant SNP content into the samples containing them. This paper is a nice survey of temporal S. pneumoniae strains within South Korea. Below I quote from the article followed by my comments which should be addressed in a revised manuscript.
“S. pneumoniae exhibits a high degree of genomic plasticity, as evidenced by the degree of genomic variability between isolates; strains typically share approximately 74% identity at the nucleotide level [10].”
This statement needs to be clarified. Many readers will assume the 74% refers to Average Nucleotide Identity (ANI) across the genome which I am sure is not the case. I believe this is a statement about what portion of the genome is shared in common between strains. Is this supposed to be the core pan-genome shared between all strains, the average amount shared between any two strains, or something else? Is this for all strains or just 19A-ST320 strains?
It would be helpful to have this same measure for the 26 isolates sequenced for this study and for the 178 strains analyzed in this study.
“To identify the genetic differences between S. pneumoniae 99-176 and the other strains, the other 25 19A-ST320 isolates were sequenced using the Illumina HiSeq 2000 Preliminary Performance Parameters (151 bp paired-end reads). Compared with those of 99-176, the mapped sites of the other 25 isolates were 99.4% to 100%. The reads were deposited in the NCBI Sequence Read Archive under the GenBank accession number PRJNA671606.”
It is not clear what the 99.4% to 100% is referring to. What is the definition of a “mapped site”? Is this based on read mapping accuracy, percentage of reads mapped, quality of calls at SNP positions, or something else?
“To map the reads to the reference, the trimmed reads for the 178 S. pneumoniae 19A-ST320 isolates, including the 25 isolates identified in this study, were aligned against the reference genome of 99-176 using Snippy version 4.4.5 (https://github.com/tseemann/snippy/). The whole-genome sequences of S. pneumoniae 19A-ST320 isolates, which were classified as global pneumococcal sequence cluster 1 (GPSC1), were retrieved from the website of The Global Pneumococcal Sequencing (GPS) Project (https://www.pneumogen.net/gps/) [15]; sequences of SP61 and SP64 were also included in the analysis. The whole-genome core SNP alignment output from Snippy was used for downstream phylogenetic analysis. A maximum-likelihood tree was generated using RAxML version 8.2.10, based on the SNPs with a minimum fraction of 90% among ≥20 reads [16]. To identify the regions of genetic recombination, we used Gubbins (v.2.3.4), which uses an algorithm to iteratively identify the loci containing elevated densities of base substitutions, while concurrently constructing a phylogeny based on the putative point mutations outside of these regions [17]. The resulting phylogenetic tree, isolate metadata, core genome SNPs, and recombination sites were visualized using Phandango version 1.3.0 [18].”
Would be good to emphasize a little more that Illumina reads for all 178 strains were used to call SNPs and not assembled contig sequences.
“The genome of S. pneumoniae 99-176, which was determined de novo as a reference, was found to be 2,089,994 bp in length, containing 2,064 coding sequences, as well as 58 tRNA and 12 rRNA genes. The sequencing statistics of the other S. pneumoniae isolates are shown in Table S1. The numbers of contigs of the isolates analyzed in this study, except for isolate 15-019, ranged from 22 to 30. A total of 237 contigs were obtained and analyzed from isolate 15-019. The assembled sequence lengths of the 24 isolates except for 15-019 ranged from 2,026,658 to 2,083,098 bp, which covered 97.0% to 99.7% of the genome of isolate 99-176. The overall G+C contents of 26 isolates were 39.52 to 39.84%.”
Why the length differences? Repeats for most? Plasmid for much longer one? Assembly methods for the Illumina reads strains were not given. The assembly method for S. pneumoniae 99-176 was not given either. Why so many contigs for 15-019, lots of IS elements or other repeats, low read coverage, contaminating sequence, intermingling of more than one strain, or something else?
“The SNPs of the isolates were determined via comparison of their whole-genome se- quences with those of isolate 99-176 (Table 1).”
I think this means the Illumina reads were used for SNP prediction but this should be clarified since assembled contigs were also generated. Did all of the contigs map to S. pneumoniae 99-176? What percentage of the reads mapped to S. pneumoniae 99-176? Were Illumina reads generated for S. pneumoniae 99-176? If so were SNPs predicted using those reads as a quality check? Were any of the positions on S. pneumoniae 99-176 consistently called as a SNP for all of the other genomes indicating a possible assembly consensus error? What size are the indels – single bp? Total size of the recombination regions would be a good extra column for Table 1.
“We analyzed the whole-genome sequences of 179 S. pneumoniae 19A-ST320 isolates, 161 including 153 isolates that belong to GPSC1.15”
I think this is a poorly formatted indication that GPSC is from reference 15.
In Figure 2A, the “genome” line at the top right with blue tick marks above and below I assume represent genes on one or the other strand but I did not see this stated anywhere and it seems unnecessary unless the figure is zoomable with gene labels. The first six “columns” before country and continent are not explained and also appear to impart little value. These six columns have been dispensed with in Figure 2B but not the “genome” line.
Author Response
- “S. pneumoniae exhibits a high degree of genomic plasticity, as evidenced by the degree of genomic variability between isolates; strains typically share approximately 74% identity at the nucleotide level [10].”
This statement needs to be clarified. Many readers will assume the 74% refers to Average Nucleotide Identity (ANI) across the genome which I am sure is not the case. I believe this is a statement about what portion of the genome is shared in common between strains. Is this supposed to be the core pan-genome shared between all strains, the average amount shared between any two strains, or something else? Is this for all strains or just 19A-ST320 strains?
- As supposed by Reviewer, we intended to state that 74% of genome is shared in common between pneumococcal strains. And it corresponds to core pan-genome shared between all pneumococcal strains. We revise the sentence.
“S. pneumoniae exhibits a high degree of genomic plasticity, as evidenced by the degree of genomic variability between isolates; it was reported that approximately 74% of genome is shared in common between strains as pan-genome [10].” (Line 55-57)
It would be helpful to have this same measure for the 26 isolates sequenced for this study and for the 178 strains analyzed in this study.
- The strain 99-176 was sequenced de novo, but the other strains were sequenced using the Illumina. The genomes of the other strains covered 99.4% to 100% of that of 99-176. Thus, it could be measured that nearly entire genome is common between strains sequenced in this study.
“The sequences of the other 25 isolates covered 99.4% to 100% of those of the strain 99-176.” (Line 94-95)
- “To identify the genetic differences between S. pneumoniae 99-176 and the other strains, the other 25 19A-ST320 isolates were sequenced using the Illumina HiSeq 2000 Preliminary Performance Parameters (151 bp paired-end reads). Compared with those of 99-176, the mapped sites of the other 25 isolates were 99.4% to 100%. The reads were deposited in the NCBI Sequence Read Archive under the GenBank accession number PRJNA671606.”
It is not clear what the 99.4% to 100% is referring to. What is the definition of a “mapped site”? Is this based on read mapping accuracy, percentage of reads mapped, quality of calls at SNP positions, or something else?
- As answered in #1, the genomes of the strains sequenced by Illumina covered 99.4% to 100% of that of 99-176 sequenced de novo. We revised the sentence.
“The sequences of the other 25 isolates covered 99.4% to 100% of those of the strain 99-176.” (Line 94-95)
- “To map the reads to the reference, the trimmed reads for the 178 S. pneumoniae 19A-ST320 isolates, including the 25 isolates identified in this study, were aligned against the reference genome of 99-176 using Snippy version 4.4.5 (https://github.com/tseemann/snippy/). The whole-genome sequences of S. pneumoniae 19A-ST320 isolates, which were classified as global pneumococcal sequence cluster 1 (GPSC1), were retrieved from the website of The Global Pneumococcal Sequencing (GPS) Project (https://www.pneumogen.net/gps/) [15]; sequences of SP61 and SP64 were also included in the analysis. The whole-genome core SNP alignment output from Snippy was used for downstream phylogenetic analysis. A maximum-likelihood tree was generated using RAxML version 8.2.10, based on the SNPs with a minimum fraction of 90% among ≥20 reads [16]. To identify the regions of genetic recombination, we used Gubbins (v.2.3.4), which uses an algorithm to iteratively identify the loci containing elevated densities of base substitutions, while concurrently constructing a phylogeny based on the putative point mutations outside of these regions [17]. The resulting phylogenetic tree, isolate metadata, core genome SNPs, and recombination sites were visualized using Phandango version 1.3.0 [18].”
Would be good to emphasize a little more that Illumina reads for all 178 strains were used to call SNPs and not assembled contig sequences.
- As suggested, we mentioned it.
“The whole genome sequences of 178 strains were used to just call SNPs and not assembled.” (Line 105)
- “The genome of S. pneumoniae 99-176, which was determined de novo as a reference, was found to be 2,089,994 bp in length, containing 2,064 coding sequences, as well as 58 tRNA and 12 rRNA genes. The sequencing statistics of the other S. pneumoniae isolates are shown in Table S1. The numbers of contigs of the isolates analyzed in this study, except for isolate 15-019, ranged from 22 to 30. A total of 237 contigs were obtained and analyzed from isolate 15-019. The assembled sequence lengths of the 24 isolates except for 15-019 ranged from 2,026,658 to 2,083,098 bp, which covered 97.0% to 99.7% of the genome of isolate 99-176. The overall G+C contents of 26 isolates were 39.52 to 39.84%.”
Why the length differences? Repeats for most? Plasmid for much longer one? Assembly methods for the Illumina reads strains were not given. The assembly method for S. pneumoniae 99-176 was not given either. Why so many contigs for 15-019, lots of IS elements or other repeats, low read coverage, contaminating sequence, intermingling of more than one strain, or something else?
- We assembled the Illumina reads by method of SPAdes0, and the reads of 99-176 by HGAP3 and Quiver.
“The genome of S. pneumoniae 99-176, which was determined de novo as a reference and was assembled de novo by method of HGAP3 and Quiver (Pacific Biosciences, Menlo Park, CA, USA), was found to be 2,089,994 bp in length, containing 2,064 coding sequences, as well as 58 tRNA and 12 rRNA genes.” (Line 117-120)
“The reads obtained by Illumina were assembled by SPAdes v3.11.1 (http://cab.spbu.ru/software/spades/).” (Line 122-123)
- For length differences, we did not analyze the reason. We could not find contamination in sequences, and supposed that IS elements or repeat accounts for the length differences.
- “The SNPs of the isolates were determined via comparison of their whole-genome sequences with those of isolate 99-176 (Table 1).”
I think this means the Illumina reads were used for SNP prediction but this should be clarified since assembled contigs were also generated. Did all of the contigs map to S. pneumoniae 99-176? What percentage of the reads mapped to S. pneumoniae 99-176? Were Illumina reads generated for S. pneumoniae 99-176? If so were SNPs predicted using those reads as a quality check? Were any of the positions on S. pneumoniae 99-176 consistently called as a SNP for all of the other genomes indicating a possible assembly consensus error? What size are the indels – single bp? Total size of the recombination regions would be a good extra column for Table 1.
- We did not assemble the sequences, but identified SNPs from Illumina reads by comparing them with whole genome sequences of 99-176, which were generated de novo by PacBio. We checked the quality of SNPs by threshold of 0.9 and depth over 30 using Snippy (v4.4.5). The positions of SNPs were represented as positions on genome of 99-176, because we did not assemble the sequences of the other 25 isolates. We identified only SNPs, that is, single bp. We think that the total size of the recombination regions is not meaningful, because the size of each recombination region differs each other and the recombination region scattered or neighboring on the genome.
“The SNPs of the isolates were determined via comparison of their Illumina reads with whole genome sequences of isolate 99-176, which were generated de novo (Table 1).” (Line 129-130)
“The quality of SNPs was checked by threshold of 0.9 and depth over 30 using Snippy.” (Line 101-102)
- “We analyzed the whole-genome sequences of 179 S. pneumoniae 19A-ST320 isolates, 161 including 153 isolates that belong to GPSC1.15”
I think this is a poorly formatted indication that GPSC is from reference 15.
- It’s our typo error. We revised it.
- In Figure 2A, the “genome” line at the top right with blue tick marks above and below I assume represent genes on one or the other strand but I did not see this stated anywhere and it seems unnecessary unless the figure is zoomable with gene labels. The first six “columns” before country and continent are not explained and also appear to impart little value. These six columns have been dispensed with in Figure 2B but not the “genome” line.
- As suggested, we deleted the “genome” line at the top in Figure 2A. In addition, we deleted the six columns, leaving only the country and continents parts.
Reviewer 2 Report
Baek et al present a genome-wide analysis of 19A-ST320 pneumococcal isolates to investigate temporal genetic changes in isolates from a single South Korean hospital over 17 years. Interestingly, isolates from South Korea do not appear to cluster together when compared among isolates from a variety of countries indicating the selected and sequenced isolates detail in this study did not disseminate via clonal spreading. The authors then identify three recombination sites where various genes involved in metabolism, virulence and antibiotic resistance. However this study is solely descriptive; no antibiotic resistance / virulence / metabolism alterations are identified in strains that have undergone recombination events. The low sample size of isolates sequenced is a limitation and only provides hints as to the importance of recombination in pneumococcal evolution.
Minor:
- Can the authors state what the depth of coverage was with respect to the genome sequencing as this is particularly important for SNP determination
- Bacterial names should be in italics, line 85 for example
- Some minors errors throughout with respect to formatting, line 81, 241, 242 for example
Author Response
Baek et al present a genome-wide analysis of 19A-ST320 pneumococcal isolates to investigate temporal genetic changes in isolates from a single South Korean hospital over 17 years. Interestingly, isolates from South Korea do not appear to cluster together when compared among isolates from a variety of countries indicating the selected and sequenced isolates detail in this study did not disseminate via clonal spreading. The authors then identify three recombination sites where various genes involved in metabolism, virulence and antibiotic resistance. However this study is solely descriptive; no antibiotic resistance / virulence / metabolism alterations are identified in strains that have undergone recombination events. The low sample size of isolates sequenced is a limitation and only provides hints as to the importance of recombination in pneumococcal evolution.
- The goal of our study was to investigate the temporal genetic changes in a species clone. Thus, the examining of antibiotic resistance, virulence, and metabolism alterations beyonds the scope of our study.
Minor
- Can the authors state what the depth of coverage was with respect to the genome sequencing as this is particularly important for SNP determination.
- We measured the depth of coverage on reference sequence of 99-176.
“The depth of coverage on reference sequence of 99-176 was 757.164 on average, ranging from 495.246 to 1753.27.” (Line 125-126)
- Bacterial names should be in italics, line 85 for example.
- We revised bacterial names in italics.
- Some minors errors throughout with respect to formatting, line 81, 241, 242 for example.
- We revised them.